# Association between Vitamin D Deficiency and Prediabetes Phenotypes: A Population-Based Study in Henan, China

**DOI:** 10.3390/nu16131979

**Published:** 2024-06-21

**Authors:** Guojie Wang, Shixian Feng, Jiying Xu, Xiaolin Wei, Guojun Yang

**Affiliations:** 1Public Health School, Henan Medical College, Zhengzhou 451191, China; 2Institute for Chronic Disease Control, Henan Provincial Center for Disease Control and Prevention, Zhengzhou 450016, China; 3Anti-TB Institute, Henan Provincial Center for Disease Control and Prevention, Zhengzhou 450016, China; 4Dalla Lana School of Public Health, University of Toronto, Toronto, ON M5S 1A1, Canada

**Keywords:** vitamin D, risk factor, impaired fasting glucose, impaired glucose tolerance

## Abstract

The evidence remains inconsistent regarding whether vitamin D deficiency (VDD) increases the risk of prediabetes. This study aimed to examine whether there is sex-specific association between VDD and impaired fasting glucose (IFG) or impaired glucose tolerance (IGT) in Henan. The data were sourced from the survey of chronic diseases and nutrition in Henan. Multinomial logistic regression models based on complex sampling design and weight were developed to estimate the odds ratio (OR) and confidence interval (95%CI) for measuring the association between VDD and IFG/IGT. The prevalence rate of IGT in men was 20.1% in the VDD group, significantly higher than that in the non-VDD group (10.5%), but no significant difference was observed in women between the VDD and non-VDD groups; there were no significant differences in IFG prevalence between the VDD and non-VDD groups in either men or women. It was found that the association between VDD and IGT was statistically significant in men. The adjusted OR (95%CI) of VDD was 1.99 (1.24–3.19) for IGT in men and 14.84 (4.14–53.20) for IGT in men having a family history of DM. Thus, men with VDD were more likely to live with IGT than those without VDD, especially for men having a family history of diabetes.

## 1. Introduction

Prediabetes is a key and critical period for the prevention and control of diabetes mellitus (DM). It is referred to as a health condition of intermediate hyperglycemia, in which the blood glucose levels do not meet the diagnosis criteria for DM but are higher than normal thresholds. It includes two phenotypes based on the definition of intermediate hyperglycemia from the WHO and Chinese Diabetes Society, impaired fasting glucose (IFG) and impaired glucose tolerance (IGT) [1,2]. Prediabetes increases the average risk for the future development of type 2 DM (T2DM) by threefold to tenfold, and approximately 70% of prediabetic persons will progress to T2DM in the future [3,4]. Besides T2DM, persons with prediabetes are at increased risk of heart attacks and strokes [4,5,6,7]. Furthermore, the prevalence of prediabetes is increasing and imposes significant public health burdens [4,7].

Vitamin D is an essential fat-soluble vitamin and plays a central role in calcium-phosphate homeostasis and skeletal health. More and more studies have reported that vitamin D has various extraskeletal functions, one of which is that vitamin D may regulate glucose homeostasis through improving glucose tolerance and insulin secretion and sensitivity [8,9,10,11,12]. Some observational and intervention studies have also indicated a significantly inverse association between vitamin D status and blood glucose level, suggesting that VDD increases the risk of DM and prediabetes, but a few studies did not find such an association [13,14,15,16,17]. The overall evidence remains inconsistent and no conclusive evidence is available, especially for prediabetes [12]. The evidence from China is limited and inconsistent as well. Three studies, including two in west China and one in east China, have been reported [18,19,20]. Of them, one study suggested that low 25(OH)D levels might independently contribute to the incidence of prediabetes or T2DM in Chinese individuals, but the other two, which both had a large sample size, did not find that lower serum 25(OH)D concentrations were associated with a higher risk of prediabetes or T2DM.

Sex differences have been observed in the association between vitamin D status and immunomodulatory and anti-inflammatory effects in some diseases [21,22], insulin resistance in a large Caucasian population [23], serum ferritin levels in Korean adults [24], and prevalence of dyslipidaemia in the middle-aged and elderly Chinese population [25]. A study in Americans over 50 years old demonstrated there was an association between VDD and prediabetes, but did not find any sex differences in the association between vitamin D status and prediabetes [26]. We were interested in whether sex differences in the association between vitamin D status and prediabetes could be distorted by the heterogeneity of prediabetes phenotypes, and whether this could be observed in the Chinese population. To our knowledge, there are no previous studies that examined sex differences in the association between VDD and prediabetes phenotypes, IFG and IGT, separately. IFG and IGT have different underlying pathophysiologic mechanisms, although both have a higher risk of developing T2DM, reflecting the impairment of blood glucose regulation function under basal state and after glucose load, respectively [3,27]. IFG is primarily caused by hepatic insulin resistance and disturbed first-phase insulin secretion, whereas IGT primarily results from muscle insulin resistance, impaired first- and second-phase insulin secretion, and reduced β-cell sensitivity to glucose. It was observed that IFG occurs more frequently in men than in women, whereas IGT is common particularly in women, which suggests that the pathophysiologic mechanisms between men and women are not completely the same [3]. It was reported that endogenous estrogens might stimulate insulin synthesis and secretion, and might be relevant the sensitivity of insulin, so female sex hormones may play an important role in the pathogenesis of IFG and IGT. Therefore, it will be more reasonable to consider the possible heterogeneity of prediabetes phenotypes and explore the association between VDD and IFG/IGT, instead of prediabetes, in men and women, respectively.

This present study aimed to examine whether there is sex-specific association between VDD and IFG/IGT after adjustment for other potential confounding factors including biological factors, lifestyles, etc., based on sampling data representative of general adult residents in Henan, in the central part of China.

## 2. Materials and Methods

### 2.1. Participants and Sampling

This present study was set in Henan, one of China’s most populous provinces in the central part of China, and the data used were sourced from the Cross-Sectional Survey of Chronic Diseases and Nutrition among the Adults in Henan, which was carried out in the period of October 2018–February 2019. The participants were selected using the multi-stage stratified cluster sampling method to represent the whole non-institutionalized population in Henan. A total of 8455 adult residents aged 18 years old or older from 84 villages or neighborhoods in 42 subdistricts or townships were selected who had completed the questionnaire survey, lab test, and vitamin D status assessment. Of them, 1774 residents who were diagnosed as DM according to the criteria mentioned below and 1054 residents who had a possible history of taking vitamin D supplements during the past month were excluded from this analysis, so finally 5627 residents selected were included as participants for this analysis.

### 2.2. Data Collection

This survey integrated three components, including a questionnaire-based interview, anthropometry, and lab test. The information collected through the questionnaire-based interview included demographic characteristics (such as sex, age, ethnicity, marital status, education attainment, occupation, etc.), family history of chronic diseases, life styles (including smoking, sedentary time, vitamin/calcium-containing nutrient supplements, etc.). Anthropometric data including weight, height, and waist circumference (WC) were measured using standard scales and procedures with participants wearing light clothes and no shoes. Body mass index (BMI) was calculated as weight (kg) divided by height squared (m^2^) (weight/height^2^). BMI ≥28 kg/m^2^ was classified as obesity and WC ≥ 90 cm in men or ≥85 cm in women were classified as abdominal obesity [2]. Blood pressure (BP) was measured with an electronic sphygmomanometer (OMRON HBP1300). Hypertension was diagnosed at ≥140/90 mmHg or as diagnosed previously in the hospital and having taken anti-hypertension medication during the past two weeks.

Blood samples were collected after an overnight fasting (at least an eight-hour fast) from an antecubital vein for measuring the concentrations of serum 25(OH)D, serum lipid profile including total cholesterol (TC), low-density lipoprotein cholesterol (LDL-C), high-density lipoprotein cholesterol (HDL-C), and triglycerides (TG), fasting plasma glucose (FPG), and 2 h plasm glucose (2 h PG) during an oral glucose tolerance test (OGTT). The OGTT was conducted with 75 g of anhydrous glucose for the participants without a history of DM after collection of the fasting blood sample, and blood samples were collected 2 h after the test load for measuring 2 h PG. All samples were tested in the lab of Guangzhou KingMed Diagnostics Group, a third-party medical test lab, except for FPG and 2 hPG during OGTT, which were performed within 48 h in the local hospitals. The TC (<4.5 mmol/L), TG (<1.7 mmol/L), HDL-C (Male > 1.0 mmol/L; Female > 1.3 mmol/L), and LDL-C (<2.6 mmol/L) were categorized in two levels for the analysis [2].

### 2.3. Vitamin D Status (VDS) Assessment

Serum 25(OH)D (25 hydroxy D) concentration was used as the indicator to assess the vitamin D status in participants. Since 25(OH)D mainly consists of 25(OH)D2 and 25(OH)D3 (25 hydroxy D2 and D3) in the blood, serum 25(OH)D concentration is defined as the total of 25(OH)D3 and 25(OH)D2 concentrations. Both were measured with high performance liquid chromatography-tandem mass spectroscopy. Standard Reference Material 972a was used as an accuracy control. Vitamin D deficiency (VDD) was defined as 25(OH)D below 12 ng/mL (30 nmol/L) [28].

### 2.4. Blood Glucose Status (BGS) Definitions and Measurements

The World Health Organization (WHO) diagnostic criteria were used to diagnose DM, IFG, and IGT [1,6]. Participants who met at any one of criteria were diagnosed with DM as follows: FPG ≥ 7.0 mmol/L; 2 h PG during OGTT ≥ 11.1 mmol/L; a self-reported diagnosis of DM that was determined previously by a healthcare professional at a hospital at the level of a town-level/community health service center or above.

Prediabetes was defined by the presence of IFG or IGT among participants without DM, which were determined based on FPG and 2 h PG during 75 g OGTT. IFG was defined as FPG between ≥6.1 and <7.0 mmol/L, and 2 h PG during 75 g OGTT levels < 7.8 mmol/L. IGT was defined as FPG < 7.0 mmol/L and 2 h PG during 75 g OGTT ≥ 7.8 and <11.1 mmol/L [1,2,6]. Normal blood glucose (NBG) was defined as neither DM nor prediabetes. Therefore, BGSs among the participants were divided into three groups, NBG, IFG, and IGT, during the analysis.

### 2.5. Statistical Analysis

The complex sample survey design and sampling weight were considered for the analysis in order to obtain unbiased estimated results representing the whole non-diabetic population ≥ 18 years old in Henan. The prevalence rates and their 95% confidence interval (95%CI) of IFG, IGT, and prediabetes in the adults were estimated, respectively, by dividing the weighted frequency of IFG, IGT, and prediabetes by the weighted total non-diabetic adults. Design-based F statistic, which was derived with non-integer degrees of freedom by using a second-order Rao and Scott correction, accounting for the survey design, from the usual Pearson Chi-square statistic for two-way tables under the survey module, was used for the comparison of rates or proportions [29].

A weighted multinomial logistic regression model with robust variance estimation of Taylor linearization based on complex sampling design was applied to estimate the odds ratio (OR) and 95% confidence interval (95%CI) for measuring the association between VDD (as the independent variable) and BGS in men, women, and both. Furthermore, multinomial logistic regression models, including the interaction terms and the stratified analyses, were conducted to identify potential effect modifiers.

All statistical analyses were performed with Stata. Stata’s survey data analysis module was employed for the estimates of prevalence rates with 95%CI and for conducting all logistic regression analyses. *p*-values less than 0.05 were considered statistically significant.

## 3. Results

### 3.1. Demographic Characteristics of Participants

Of the 5627 participants, men and women accounted for 45.5% (2558/5627) and 54.5% (3069/5627), respectively; 58.3% (3279/5627) were aged between 40–64 years; and 98.7% (5552/5627) were of Han ethnicity. The demographic characteristics of participants are presented in Table 1.

### 3.2. BGS and Vitamin D Status among Participants

Of the 5627 participants, 635 (11.3%), including 346 (13.5%) men and 289 (9.4%) women, lived with IFG; 1064 (18.9%), including 438 (17.1%) men and 626 (20.4%) women, lived with IGT; and 1699 (30.2%), including 784 (30.6%) men and 915 (29.8%) women, lived with prediabetes. The proportion of VDD was 24.0% across all participants, with 14.7% in men and 31.8% in women. (Table 2)

### 3.3. Estimated Prevalence Rates of IFG, IGT, and Prediabetes by Vitamin D Status and Sex in Adults Based on the Sampling Weight

Overall, the prevalence rate of IFG in the adults was 11.8%, and it was 9.7% in the VDD group and 12.5% in the non-VDD group, having no significant difference (χ^2^ = 7.59, design-based F = 1.63, *p* = 0.226), and after being stratified by sex, the prevalence rates of IFG between the VDD and non-VDD groups had no significant differences in either men or women (both *p* > 0.05). Overall, the prevalence rate of IGT in the adults was 14.8%, and it was 20.1% in men with VDD and 10.5% in men with non-VDD, having a statistically significant difference (χ^2^ = 34.26, design-based F = 5.58, *p* = 0.036). The prevalence rates of IGT in women with VDD and with non-VDD were 14.2% and 19.7%, respectively, and the difference was not significant (χ^2^ = 14.09, design-based F = 2.01, *p* = 0.182). The prevalence rate of prediabetes was estimated to be 26.6% (95%CI: 18.9–36.1%) in the adults. It was 26.1% in the adults with VDD and 26.8% in the adults with non-VDD, and the difference was not statistically significant (χ^2^ = 0.217, design-based F = 0.03, *p* = 0.863). (Table 3)

### 3.4. The Association between Vitamin D Status and IFG/IGT

Taking blood glucose status (BGS) as the dependent variable and NBG category of BGS as the base outcome, multinomial logistic regression analysis adjusted all variables studied showed that VDD was not significantly associated with IFG/IGT in the adults. However, the multinomial logistic regression analysis including the interaction term between VDD and sex in the adults demonstrated that there was an interaction between VDD and sex for the risk of IGT, and the OR of the interaction term between VDD and sex was estimated as 0.39 (0.18–0.84) for IGT with statistical significance (*p* = 0.020), suggesting that the effect of VDD on the risk of IGT differed with sex; no significant interaction was found for IFG. After stratification by sex, the analysis results of multinomial logistic regression adjusted for various confounding variables studied (Model 1–4) all showed that the association between VDD and IFG was not statistically significant either in men or women (all *p* > 0.05), while the association between VDD and IGT was statistically significant in men (all *p* < 0.05) but not in women (all *p* > 0.05), suggesting that the results of the four models including different numbers of potential confounding variables were consistent and the association was stable. The adjusted OR (95%CI) of VDD estimated in the model with adjustment for all variables studied was 1.99 (1.24–3.19) for IGT in men and 0.79 (0.43–1.43) in women, implying that VDD was the independent risk factor of IGT and doubly increased the risk of IGT in men, but not in women. No significant association between VDD and IFG was found either in men or women (Table 4 and Table 5).

In addition, it was found that besides VDD and sex, age, education, marital status, alcohol consumption, hypertension, and TC were significantly associated with IGT, but ethnicity, family history of DM, occupation, residential location, smoking status, sedentary time, test season, abdominal obesity, obesity, TG, HDL-C, and LDL-C were not; sex, age, education attainment, occupation, TC, and LDL-C were significantly associated with IFG, but ethnicity, family history of DM, marital status, residential location, alcohol consumption, smoking status, sedentary time, test season, abdominal obesity, obesity, hypertension, TG, and HDL-C were not. Although the analysis results showed that ethnicity, test season, and obesity were neither significantly associated with IGT nor IFG based on the significance level of *p* < 0.05, perhaps it was the sample size that resulted in the insignificant associations between these three factors and IGT/IFG (Table 4).

The interactions between VDD and the other independent variables were examined, and it was found that family history of DM was probably an effect modifier for the association between VDD and IGT in men, and no other potential effect modifiers were identified. After stratification by family history of DM, the OR (95%CI) of VDD adjusted for all studied variables was estimated to be 14.84 (4.14–53.20) for IGT in men having a family history of DM and 1.29 (0.73–2.28) in men with having no family history of DM, respectively. By comparison, the adjusted OR (95%CI) of VDD was estimated to be 1.10 (0.18–6.71) and 1.02 (0.50–2.09) for IFG in men having a family history of DM and having no family history of DM, respectively (Table 6).

## 4. Discussion

This present study found that the prevalence rate of IGT in men with VDD was 20.1%, significantly higher than that in men with non-VDD (10.5%), and that there were effect modifications by sex on the association between VDD and IGT in adults and by the family history of DM in men, suggesting VDD was independently associated with IGT in men, especially in men having a family history of DM. The OR (95%CI) of VDD adjusted for all studied variables was estimated to be 1.99 (1.24–3.19) for IGT in men and 14.84 (4.14–53.20) for IGT in men having a family history of DM, implying that after excluding the impact of other factors, having VDD doubly increased the risk of having IGT in men compared with men with non-VDD and increased by nearly 15 times the risk of having IGT in men with a family history of DM. It seems men having a family history of DM living with VDD are probably far more likely to have IGT than those who are not living with VDD, and that the association between VDD and IGT mainly exists in men with a positive family history of DM. Meantime, no independent associations between vitamin D status and prevalence of IFG were found either in men or women.

We did not find any previous sex-specific studies that examined the association between VDD and IGT/IFG, only prediabetes. For comparison, a previous study based on the data from the cross-sectional surveys of the NHANES in the USA reported that VDD was associated with risk of prediabetes in Americans over 50 years old, but there was no effect modification by sex and it did not report an association between vitamin D status and the phenotypes of prediabetes [26]. However, studies based on the Korean National Health and Nutrition Examination Survey (KNHANES) data reported that there was a sex-based modification effect of vitamin D on blood glucose regulation and prevalence of metabolic syndrome, and also suggested that the association between vitamin D status and diabetes or metabolic syndrome was only found in men [30,31]. One Chinese study reported that vitamin D had a negative impact on insulin resistance only in male patients with newly diagnosed type 2 diabetes mellitus [32]. Although these studies in Korea and China did not directly explore the association between vitamin D and IFG/IGT, they did demonstrate that the blood glucose regulation function of vitamin D perhaps only existed in men. It has been reported there is a potential interaction between sex hormones and level of vitamin D, and estrogen can enhance some functions of vitamin D [21,22]. To a certain extent, these findings support that the association between VDD and BGS might be sex-differentiated, if the association exists. This is in line with our findings that the association between VDD and IGT was only found in men, but not in women. However, this seems inconsistent with the study in USA mentioned above. Perhaps racial differences and dietary structure disparities lead to this inconsistency.

Additionally, a further examination on the association between VDD and IGT in men was conducted in the present study by adding the interaction term between VDD and other factors to the logistic regression model, finding that there might be an interaction between vitamin D status and family history of DM, showing a far stronger association between VDD and IGT in men having a family history of DM, suggesting that VDD greatly increased the risk of IGT in men having a family history of DM. The population having a family history of DM is a high-risk group for IGT since a positive family history of DM increases the risk of IGT by 25–64% [33]. It is reasonable that the association between VDD and IGT is more likely to be demonstrated in the high-risk groups of IGT, such as men having a family history of DM. On the other hand, this finding suggests that the occurrence of IGT may be effectively reduced by preventing or treating VDD in men having a family history of DM, but this needs confirmation using clinical experiments.

This present study did not find any independent association between VDD and IFG either in men or women. IFG and IGT are two different phenotypes of prediabetes and both increase the risks of progressing to DM, but they have different underlying pathophysiological mechanisms, reflecting the impairment of blood glucose regulation function under basal state and after glucose load, respectively [3,27]. Owing to such heterogeneity in prediabetes phenotypes, the risk factors for them should not be completely the same. Thus, when exploring the association between vitamin D status and prediabetes, it was more reasonable to take IFG and IGT into account separately in this present study. Additionally, this drops a hint that it may be more reasonable to use IFG and IGT instead of prediabetes for the purposes of the diagnosis, clinical intervention, prevention, and control of DM in men and women.

Currently, there is no international consensus for the definitions of prediabetes, DM, and VDD [1,2,4,7,11,17,28]. The prevalent definitions or diagnosis criteria of DM, prediabetes, and its phenotypes issued from the WHO, IDF, and ADA are used with different biomarkers, including glycated hemoglobin (HbA1c) as well as FPG and 2 h PG during OGTT, or different cut-off values for same biomarkers, all of which provide different estimates on the prevalences of DM, prediabetes, and its phenotypes in the population [4,7]. Considering that HbA1c being used for the definition of prediabetes has not been universally adopted globally, and that the current clinical practice in Henan for HbA1c testing has not been put into use for the diagnosis of prediabetes and diabetes in all healthcare institutions, this present study adopted the definition of prediabetes issued from the WHO and Chinese Diabetes Society, which does not include the biomarker of HbA1c [1,2,6,7]. The definition of VDD is under debate as well. The cut-off values of serum 25(OH)D concentration for the definition of VDD currently used include 10 ng/mL, 12 ng/mL, 20 ng/mL, and 30 ng/mL [11,17,28]. The working group convened by the Sackler Institute for Nutrition Science at the New York Academy of Sciences and the Bill & Melinda Gates Foundation in coordination with a scientific organizing committee conducted a study and recommended the cut-off value 12 ng/mL (30 nmol/L) was for low- and middle-income countries [28]. The cut-off value of 12 ng/mL was also recommended for the public health approach. Considering the income of Henan residents, this present study took 12 ng/mL as the cut-off value for the definition of VDD. The difference in the definitions or criteria would bring about the bias on the association between VDD and prediabetes or its phenotypes, which may be a factor leading to the inconsistent study results reported on the association between VDD and prediabetes besides the sex modification effect and heterogeneity of prediabetes phenotypes.

This present study considered the heterogeneity of prediabetes phenotypes as well as the effect modification by sex to examine the association between VDD and IFG/IGT based on a population-representative cross-sectional survey with a multi-stage sampling method, and statistical analysis methods for complex sample survey data were used incorporating both the sampling weights and the sample design instead of the estimate methods based on a simple random sample, which produced unbiased estimates and the correct standard error estimates. Therefore, the estimates in this present study are reliable. However, this present study has some limits as well. HbA1c was not included as a phenotype of prediabetes, which would result in the underestimated prevalence of prediabetes, but had no impact on the conclusion of this present study since this present study focused on the two phenotypes of prediabetes, IFG and IGT. This is a cross-sectional study design which cannot draw causal relationships, owing to the nature of cross-sectional studies, which lack temporality.

## 5. Conclusions

In conclusion, this present study, a population-based study conducted in a middle-income region with a warm temperate–tropical climate, Henan, in the central part of China, found there was a sex disparity in the association between VDD and IGT, a phenotype of prediabetes: VDD was independently associated with IGT in men, but not in women; men with VDD were more likely to have IGT than those without VDD, and the risk differed with family history of DM, having a far stronger association in men having a positive family history of DM. It is suggested that future perspective cohort studies or trials for examining the causal relationship between VDD and prediabetes consider the modification effect of sex and family history of DM, and the heterogeneity of prediabetes phenotypes.

## Figures and Tables

**Table 1 nutrients-16-01979-t001:** Demographics of participants by sex [n (%)].

Demographics	Men		Women		Both	
	n	%	n	%	n	%
Age (years)						
18–39	456	17.8	601	19.6	1057	18.8
40–64	1454	56.8	1825	59.5	3279	58.3
65–104	648	25.4	643	21.0	1291	22.9
Ethnicity						
Han	2528	98.8	3024	98.5	5552	98.7
Other	30	1.2	45	1.5	75	1.3
Marital status						
Married	2372	92.7	2851	92.9	5223	92.8
Single *	186	7.3	218	7.1	404	7.2
Occupation						
AFAHFWRI *	916	35.8	918	29.9	1834	32.6
Non-AFAHFWRI	899	35.1	635	20.7	1534	27.3
Unemployed/housework	457	17.9	1207	39.3	1664	29.6
Retirement	286	11.2	309	10.1	595	10.6
Education attainment						
Primary school or below	705	27.6	1343	43.8	2048	36.4
Junior high school	1088	42.5	1039	33.9	2127	37.8
SHS/TSS *	562	22.0	469	15.3	1031	18.3
College or above	203	7.9	218	7.1	421	7.5
Residential location						
Urban area	1301	50.9	1687	55.0	2988	53.1
Rural area	1257	49.1	1382	45.0	2639	46.9

* Single includes persons who have never been married, are in divorce, or have lost their spouses; AFAHFWRI and SHS/TSS are short for agricultural, forestry, animal husbandry, fishery and water resource industries and senior high school/technical secondary school, respectively.

**Table 2 nutrients-16-01979-t002:** Blood glucose status and vitamin D status among the participants by sex.

	Men		Women		Both	
	n	%	n	%	n	%
BGS						
NBG	1774	69.4	2154	70.2	3928	69.8
IFG	346	13.5	289	9.4	635	11.3
IGT	438	17.1	626	20.4	1064	18.9
Prediabetes	784	30.6	915	29.8	1699	30.2
Vitamin D status						
VDD	376	14.7	977	31.8	1353	24.0
Non-VDD	2182	85.3	2092	68.2	4274	76.0

**Table 3 nutrients-16-01979-t003:** Estimated prevalence rates of IFG, IGT, and prediabetes by vitamin D status and sex (%, 95%CI).

BGS	Vitamin D Status	Men	Women	Both
IFG	VDD	13.0 (6.4–24.9)	8.1 (4.4–14.4)	9.7 (5.4–17.0)
	Non-VDD	15.7 (9.5–25.0)	8.0 (5.3–11.9)	12.5 (7.9–19.3)
	Total	15.3 (9.3–24.1)	8.0 (5.1–12.3)	11.8 (7.4–18.3)
	χ^2^, design-based F, P	1.86, 0.42, 0.527	0.01, 0.005, 0.947	7.59, 1.63, 0.226
IGT	VDD	20.1 (9.3–40.0)	14.2 (9.0–21.7)	16.4 (10.6–24.6)
	Non-VDD	10.5 (8.5–12.8)	19.7 (13.5–27.6)	14.3 (11.0–18.4)
	Total	12.1 (8.5–16.9)	17.8 (12.8–24.2)	14.8 (11.3–19.2)
	χ^2^, design-based F, P	34.26, 5.58, 0.036	14.09, 2.01, 0.182	3.70, 0.72, 0.414
Prediabetes	VDD	33.9 (21.8–48.5)	22.3 (13.5–34.4)	26.1 (17.3–37.4)
	Non-VDD	26.2 (18.5–35.7)	27.6 (18.8–38.6)	26.8 (18.8–36.6)
	Total	27.4 (19.6–36.9)	25.8 (17.9–35.7)	26.6 (18.9–36.1)
	χ^2^, design-based F, P	10.10, 2.79, 0.121	10.36, 1.13, 0.309	0.22, 0.03, 0.863

**Table 4 nutrients-16-01979-t004:** Analysis results of multivariate multinomial logistic regression including all studied variables and the interaction term between VDD and sex.

Variable	IFG			IGT		
	OR (95%CI)	t	*p* > t	OR (95%CI)	t	*p* > t
VDD						
Yes	0.98 (0.49–2.00)	−0.05	0.963	1.95 (1.12–3.42)	2.61	0.023
No	1.00			1.00		
Sex						
M	1.00			1.00		
F	0.52 (0.30–0.90)	−2.60	0.023	3.27 (1.59–6.73)	3.58	0.004
VDD # Sex *						
VDD = No, Sex = M	1.00			1.00		
VDD = Yes, Sex = F	1.09 (0.66–1.79)	0.38	0.713	0.39 (0.18–0.84)	−2.67	0.020
Age (years)						
18–39	1.00			1.00		
40–64	1.10 (0.67–1.81)	0.42	0.685	1.90 (1.25–2.89)	3.32	0.006
65–104	1.76 (1.11–2.79)	2.65	0.021	4.49 (2.97–6.80)	7.89	0.000
Ethnicity						
Han	1.00			1.00		
Other	0.68 (0.26–1.79)	−0.87	0.401	0.59 (0.32–1.07)	−1.93	0.078
Family history of DM *						
No	1.00			1.00		
Yes	0.78 (0.33–1.83)	−0.63	0.539	1.32 (0.70–2.48)	0.95	0.362
Education attainment						
Primary school or below	1.00			1.00		
Junior high school	1.00 (0.80–1.26)	0.01	0.991	1.06 (0.76–1.48)	0.38	0.712
SHS/TSS	0.81 (0.51–1.28)	−1.01	0.334	0.60 (0.38–0.97)	−2.31	0.040
College or above	0.32 (0.12–0.81)	−2.67	0.021	0.84 (0.51–1.37)	−0.78	0.450
Marital status						
Married	1.00			1.00		
Single	0.78 (0.48–1.28)	−1.09	0.296	1.96 (1.06–3.62)	2.38	0.035
Occupation						
AFAHFWRI	1.00			1.00		
Non-AFAHFWRI	1.73 (1.07–2.80)	2.48	0.029	1.06 (0.68–1.63)	0.27	0.791
Unemployed/housework	0.70 (0.24–2.00)	−0.75	0.470	0.80 (0.57–1.10)	−1.52	0.153
Retirement	0.35 (0.17–0.73)	−3.11	0.009	0.81 (0.46–1.43)	−0.81	0.436
Residential location						
Urban area	1.00			1.00		
Rural area	0.91 (0.27–3.06)	−0.18	0.862	0.94 (0.49–1.81)	−0.19	0.849
Alcohol consumption						
No	1.00			1.00		
Yes, ≤30 days	0.98 (0.52–1.83)	−0.09	0.933	1.58 (1.05–2.38)	2.42	0.032
Yes, >30 days	0.67 (0.37–1.23)	−1.44	0.176	0.78 (0.53–1.16)	−1.35	0.201
Smoking status						
Never	1.00			1.00		
Former	0.68 (0.38–1.24)	−1.39	0.189	1.72 (0.83–3.56)	1.63	0.129
Current	1.11 (0.60–2.03)	0.36	0.725	1.42 (0.80–2.53)	1.34	0.205
Sedentary time *						
Low	1.00			1.00		
High	0.86 (0.57–1.28)	−0.84	0.415	0.96 (0.64–1.46)	−0.20	0.847
Test season						
Fall	1.00			1.00		
Winter	1.57 (0.41–5.99)	0.73	0.481	1.87 (0.80–4.39)	1.61	0.134
Abdominal obesity						
No	1.00			1.00		
Yes	1.19 (0.54–2.60)	0.48	0.638	0.85 (0.50–1.45)	−0.66	0.520
Obesity						
No	1.00			1.00		
Yes	1.15 (0.75–1.77)	0.69	0.500	1.63 (0.90–2.96)	1.79	0.098
Hypertension						
No	1.00			1.00		
Yes	1.05 (0.83–1.33)	0.49	0.632	1.54 (1.10–2.16)	2.78	0.017
TC						
Low	1.00			1.00		
High	2.74 (1.29–5.83)	2.92	0.013	1.58 (1.11–2.25)	2.80	0.016
TG						
Low	1.00			1.00		
High	0.95 (0.45–2.00)	−0.14	0.888	1.48 (0.88–2.49)	1.64	0.127
HDL-C						
Low	1.00			1.00		
High	0.81 (0.60–1.08)	−1.60	0.136	1.08 (0.65–1.79)	0.31	0.759
LDL-C						
Low	1.00			1.00		
High	0.53 (0.33–0.86)	−2.87	0.014	0.86 (0.45–1.64)	−0.52	0.613
_cons	0.16 (0.03–0.91)	−2.30	0.040	0.02 (0.01–0.10)	−5.72	0.000

* VDD # Sex is the interaction term between VDD and Sex; sedentary time was categorized into low and high levels based on median sedentary time; having a family history of DM refers to self-reported DM in siblings, parents, or grandparents.

**Table 5 nutrients-16-01979-t005:** ORs (95%CI) estimated with multinomial logistic regression by sex for the association between VDD and IFG/IGT.

Model *	Men			Women			Both		
	OR (95%CI)	t	P	OR (95%CI)	t	*p*	OR (95%CI)	t	*p*
Model 1									
IFG	0.93 (0.49–1.76)	−0.26	0.800	0.94 (0.55–1.61)	−0.24	0.817	0.77 (0.46–1.29)	−1.10	0.292
IGT	2.22 (1.07–4.61)	2.38	0.035	0.67 (0.35–1.28)	−1.34	0.206	1.14 (0.73–1.78)	0.63	0.539
Model 2									
IFG	0.95 (0.49–1.86)	−0.16	0.873	1.05 (0.58–1.90)	0.17	0.865	0.97 (0.53–1.75)	−0.12	0.904
IGT	2.39 (1.27–4.51)	2.99	0.011	0.76 (0.36–1.61)	−0.79	0.443	1.15 (0.65–2.03)	0.53	0.606
Model 3									
IFG	0.99 (0.46–2.13)	−0.02	0.983	1.18 (0.71–1.94)	0.71	0.492	1.02 (0.52–2.01)	0.06	0.951
IGT	2.12 (1.31–3.44)	3.39	0.005	0.77 (0.40–1.46)	−0.90	0.388	1.09 (0.64–1.84)	0.34	0.738
Model 4									
IFG	0.92 (0.45–1.86)	−0.27	0.795	1.19 (0.71–2.01)	0.76	0.463	1.01 (0.53–1.91)	0.02	0.984
IGT	1.99 (1.24–3.19)	3.16	0.008	0.79 (0.43–1.43)	−0.88	0.397	1.09 (0.66–1.79)	0.36	0.725

* NBG category was taken as the base outcome (reference) group for all models. Model 1: unadjusted; Model 2: adjusted for age, ethnicity, and family history of DM in men and women sub-populations, plus sex in the whole population combined with men and women; Model 3: adjusted for the same variables in Model 3 plus education attainment, marriage status, occupation, residential location, smoking status, alcohol consumption, abdominal obesity, obesity, sedentary time, test season, and hypertension in men and women sub-populations, plus sex in the whole population combined with men and women; Model 4: adjusted for the same variables in Model 3 plus TC, TG, HDL-C, and LDL-C in men and women sub-populations, plus sex in the whole population combined with men and women.

**Table 6 nutrients-16-01979-t006:** ORs (95%CI) estimated with multinomial logistic regression by family history of DM for the association between VDD and IFG/IGT in men.

Model *	Men Having Family History of DM			Men Having No Family History of DM		
	OR (95%CI)	t	*P*	OR (95%CI)	t	*p*
Model 5						
IFG	0.40 (0.10–1.65)	−1.41	0.185	1.07 (0.55–2.09)	0.21	0.834
IGT	4.37 (1.47–13.04)	2.94	0.012	1.43 (0.70–2.91)	1.08	0.299
Model 6						
IFG	1.10 (0.18–6.71)	0.12	0.909	1.02 (0.50–2.09)	0.07	0.947
IGT	14.84 (4.14–53.20)	4.60	0.001	1.29 (0.73–2.28)	0.99	0.343

* NBG category was taken as the base outcome (reference) group for all models. Model 5: unadjusted; Model 6: adjusted for age, education attainment, marriage status, occupation, residential location, smoking status, alcohol consumption, abdominal obesity, obesity, sedentary time, test season, hypertension, TC, TG, HDL-C, and LDL-C in men having a family history of DM and having no family history of DM. Ethnicity was not included in the models since the proportion of men with non-Han ethnicity was too small.

## Data Availability

The data that support the findings of this present study are available from the corresponding author, G.W, upon reasonable request. The data are not publicly available due to their containing information that could compromise the privacy of research participants and institutional data policy.

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
