# Peer review of "Association between Vitamin D Deficiency and Prediabetes Phenotypes: A Population-Based Study in Henan, China"

_nutrients, 2024, doi:10.3390/nu16131979_

Round 1

Reviewer 1 Report

Comments and Suggestions for Authors

The manuscript entitled "Association between Vitamin D Deficiency and Prediabetes Phenotypes: A Population-Based Study in Henan, China" needs significant changes before publication. I found numerous weaknesses of the paper. Please address to my comments:

1. What do you wanted to express using "a bidirectional MR approach..." in lines 52-54 ref. 20 in the Introduction

2. The majority of studies you cited to show associations between the risk of prediabetes (or diabetes) and vitamin D deficiency is not adjusted to sex. You also wrote" To our knowledge, there were no previous studies to examine sex differences on the association between VDD and prediabetes phenotypes, IFG and IGT, separately. IFG and IGT have different underlying pathophysiologic mechanisms although both have a higher risk of developing T2DM [3, 27]." You did not provide any pathophysiologic mechanisms of IFG and IGT development in the Introduction. Are the mechanisms different between men and women? It is not clear why you decided to perform analysis between IFG and IGT and VDD adjusted to sex.

 3. There is no clear justification why you perfomed analysis according to: "Therefore, it will be better to consider the possible heterogeneity of prediabetes phenotypes and explore the association between VDD and IFG/IGT, instead of prediabetes, in men and women, respectively."  Therefore, my suggested presentation of the results would show whether the relationships suggested by the authors actually exist.

4. It is serious drawback that prediabetes was not diagnosed without HbA1c%. Are you able to assess what error you made when qualifying participants for the study without HBA1c % measurement (bias related to the allocation of participants to groups )? 

5. This data: "1699 (30.2%), including 784 (30.6%) men and 915 (29.8%) women, lived  with prediabetes." (lines164-165) is not showed in Table 2.

6. You performed also statistical analysis including "ther potential confounding factors including biological factors, lifestyles" (Tab 4), but these results are not commented in the  result section. You obtained lots of intersting and significant data  (please see age, marital status, alcohol consumption, smoking, hypertension,LDL-C). 

7. Where can the reader  find results described in this sentence "After stratification by family history of DM, OR (95%CI) of VDD adjusted for all studied variables were estimated to 14.84 (4.14-53.20) for IGT in men having family history of DM, and 1.02 (0.50-2.09) in men with having no family history of DM, respectively" lines 203-205?

8. It is difficult to find data presented in lines 188-193 in Tab 5 ("However, the multinomial logistic regression analysis including interaction term between VDD and sex in the adults demonstrated that there was interaction  between VDD and sex on the risk of IGT, and OR of the interaction term between VDD  and sex was estimated as 0.39 (0.18-0.84) for IGT with statistical significance (P=0.020),  suggesting that the effect of VDD on the risk of IGT differed with sex; no significant interaction was found for IFG. (for details see table 5)"). There is no comment about results obtained for MODEL 1-4.

9. Authors should avoid using informal language, like in lines 45, 51, 59, 237,241, 248, 267, 294  

10. What is a clinical significance of your findings " ("lines 232-236) Therefore, men having family history of DM living with VDD probably are far more likely to have IGT than those without living with VDD, and it seems that the association between VDD and IGT mainly existed in men with positive family history of DM, and no independent associations between vitamin D status and prevalence of IFG were found either in men or women")

11. Please explain precisely why women need lower level of vitamin D as you wrote in sentence  "It seems that women need lower levels of vitamin D for a same normal physiological function than men due to the difference of sex  hormones. To a certain extent these findings support that the association between VDD and BGS might be sex-different if the association exists."

12. In the paragraph describing the results of logistic regression model, authors commented only family history of DM, other confounding factors have been omitted.  

13. Could you explain more precisely what did you mean in lines 275-284. what kind of difference in the definitions did you mean?

 14. Based on the obtained results, could you say that it is reasonable to diagnose IFG or IGT instead of prediabetes in men and women? What are other clinical significances of the obtained results?

Comments on the Quality of English Language

Moderate editing of English language required

Author Response

Dear Sir or Madam:

We appreciate that you have been spending time to review our manuscript. The comments are great and very helpful, and we benefit a lot from the comments. According to comments, we have revised the manuscription very carefully. Some results, comments, or explanations requested have been added; something unnecessary or unclear have been removed, changed, modified, or polished. All revisions have been highlighted in red in the manuscription. All references are relevant to the contents of the manuscript have been checked, and one new reference have been added, the order of some reference was adjusted, and several unnecessary references have been removed, so the reference index numbers have been changed. Some gramma errors in the sentences have been corrected. 

Attached is the reponding file for every comment. We appreciate your consideration. Please let us know if you have any questions. Thank you so much.

Best regards,

Guojie Wang

Reviewer 2 Report

Comments and Suggestions for Authors

Interesting study, I just have a few comments.

1. why was a cutoff value of 12 ng / ml chosen for VDD? Most of the literature suggests 20 ng /ml. At 12 ng / ml, serere VDD should be considered. Please acknowladge and discuss. (See Nutrients. 2023 Jan 30;15(3):695. doi: 10.3390/nu15030695. )
2. Have you tried correlations between vitamin D levels and markers of pre-diabetes? Maybe you could plot it and show it in graphs?
Maybe you could also calculate the cutoff value for men and women?
3. discuss why there is no difference in women, maybe diet, habits, substance abuse?  Maybe western vs traditional Chinese diet is the differentiator (in comparison to US studies?)?
4. Have you tried comparing 12 ng/mL vs 30 ng/mL (optimal)?

Comments on the Quality of English Language

Please proofread once again

Author Response

Dear Sir or Madam

We appreciate that you have been spending time to review our manuscript. The comments are great and very helpful, and we benefit a lot from the comments in this project. According to comments, we have revised the manuscription very carefully. Some results, comments, or explanations requested have been added; something unnecessary or unclear have been removed, changed, modified, or polished. All revisions have been highlighted in red in the manuscription. All references are relevant to the contents of the manuscript have been checked, and the reference recommended have been reviewed and added to the manuscription. The order for some references were adjusted, and several unnecessary references have been removed, so the reference index numbers have been changed. In addition, some gramma errors in the sentences have been corrected.

Attached is the file for responding the comments. We appreciate your consideration. Please let us know if you have questions. Thank you so much.

Best regards,

Guojie Wang

Round 2

Reviewer 1 Report

Comments and Suggestions for Authors

Dear Authors,

 Thank you for responses to my comments.However I found that some were missed. Please look closer to my previous comment 2. You still have not provided any  mechanisms of IFG and IGT development in men and women in the Introduction.

In case of comment 6 your answer is very superficial "In addition, it was found that besides VDD and sex, age, education, marital status, alcohol consumption, hypertension and TC were associated with IGT, and sex, age, education attainment, occupation, TC and LDL-C were associated with IFG." I would like to read which confounding factor is or is not significantly associated with IFG/IGT. And which associations were or were not expected/surprising.  You missed obesity, abdominal obesity, AFAHFWRI, Retirement, and others.

 In the response to last comment (14) you wrote "Since this present study is a cross-sectional study of epidemiology, not a clinical study, we think the results can’t be used for the assessment on the diagnosis of IFG, IGT and prediabetes." Please nore that clinical trials are classified into observational and intervetion studies. A cross-sectional study belongs to observational clinical trial. You also use informal language "can't"

Comments on the Quality of English Language

Minor editing of English language required

Author Response

Dear Sir or Madam,

We appreciate that you spent time reviewing our revised manuscription, the comments are very helpful.

For comment 2,the sentences below were added to the discussion (line 317-320).

IFG and IGT have different underlying pathophysiologic mechanisms although both have a higher risk of developing T2DM, reflecting the impairment of blood glucose regulation function under basal state and after glucose load, respectively [3, 27]. IFG is primarily caused by hepatic insulin resistance and disturbed first-phase insulin secretion, whereas IGT primarily resulted from muscle insulin resistance, impaired first- and second-phase insulin secretion, and reduced β-cell sensitivity to glucose. It was observed that IFG occurs more frequently in men than in women, whereas IGT is common particularly in women, which suggests that the pathophysiologic mechanisms between men and women do not completely same [3]. It was reported that endogenous oestrogens might stimulate insulin synthesis and secretion, and might be relevant the sensitivity of insulin, so female sex hormones may play an important role in the pathogenesis of IFG and IGT.

For comment 6, the sentences below were added to the discussion (line 317-320).

In addition, it was found that besides VDD and sex, age, education, marital status, alcohol consumption, hypertension and TC were significantly associated with IGT, but ethnicity, family history of DM, occupation, residential location, smoking status, sedentary time, test season, abdominal obesity, obesity, TG, HDL-C, and LDL-C were not; sex, age, education attainment, occupation, TC and LDL-C were significantly associated with IFG, but ethnicity, family history of DM, marital status, residential location, alcohol consumption, smoking status, sedentary time, test season, abdominal obesity, obesity, hypertension, TG, and HDL-C were not. Although the analysis results showed that ethnicity, test season, and obesity were neither significantly associated with IGT nor IFG based on the significance level of P<0.05, perhaps it was the sample size that resulted in the insignificant as-sociation between three factors and IGT/IFG.  (table 4)

For comment 14, the sentences below were added to the discussion (line 317-320).

Also, it drops a hint that it may be more reasonable to use IFG and IGT instead of predia-betes for the purposes of diagnosis, clinical intervention, prevention and control of DM in men and women.

Please let us know if you have questions. Thanks so much 

Best regards,

Guojie Wang